🔓 | **Open Peer Review** | Antimicrobial Chemotherapy | Research Article

# Survival mechanism of pancreatic tumor bacteria and their ability to metabolize chemotherapy drugs

Zara Ahmad Khan,[1] Michał J. Sobkowiak,[1,2] Mahin Ghorbani,[1] Tajpara Poojabahen,[1] Khaled Al-Manei,[1,3] Asif Halimi,[4] Zeeshan Ateeb,[5,6] Volkan Özenci,[7,8] Rainer Heuchel,[9] Matthias Löhr,[9] Margaret Sällberg Chen[1,2,10]

**ABSTRACT** Pancreatic cancer (PC) remains one of the most lethal malignancies, with limited treatment efficacy. While surgical resection is the most effective option, chemotherapy with agents such as 5-fluorouracil (5-FU) and gemcitabine may improve survival. Intraductal papillary mucinous neoplasms (IPMNs) are pancreatic cystic tumors and important precursor lesions frequently detected during PC screening. Emerging evidence suggests that IPMNs can harbor a distinct tumor microbiome, but the microbial persistence and potential influence on cancer treatment remain poorly understood. In this study, we analyzed bacterial isolates from clinical IPMN samples and investigated their interactions with chemotherapeutic agents using functional assays and whole-genome sequencing (WGS). We found that most isolates reduced the cytotoxic effect of 5-FU and gemcitabine on pancreatic cancer cell lines (PANC-1, AsPC-1, and Capan-2). WGS revealed that the Gammaproteobacteria strains were enriched in genes associated with antibiotic resistance, drug transport, and virulence compared to the Bacilli strains. Further pathway analysis showed that Gammaproteobacteria were enriched in pyrimidine metabolism pathways, while Bacilli were enriched in purine metabolism. These findings indicate that IPMN-associated bacteria are metabolically active and capable of modulating chemotherapy drug efficacy. Together, our findings suggest that the microbial adaptation mechanisms supporting bacterial survival within tumor lesions also enable them to interact with pyrimidine analogs. This underscores the importance of elucidating the functional roles of tumor-associated microbiota in modulating the tumor microenvironment and treatment efficacy.

**IMPORTANCE** Chemotherapy is a primary treatment for pancreatic cancer, and emerging evidence indicates that the gut microbiota can modulate its efficacy. While most studies have focused on gut-residing microbes, characterization of intra-tumoral microbes within the pancreas remains limited. Here, we report new insights into metabolic interactions between chemotherapeutic agents and patient-derived pancreatic tumor bacteria. These bacteria were isolated from intraductal papillary mucinous neoplasms (IPMNs), the main precursors to pancreatic cancer. Our findings demonstrate that patient-derived pancreatic tumor bacteria tolerate two commonly used chemotherapeutic drugs, 5-fluorouracil (5-FU) and gemcitabine, and can attenuate their cytotoxic effects on pancreatic cancer cells. Through whole-genome and transcriptomic analyses, we further reveal potential adaptation mechanisms that could enable these bacteria to persist in the tumor microenvironment and metabolize chemotherapeutics.

**KEYWORDS** pancreatic neoplasm, IPMN, chemotherapeutics, tumor microbiome, WGS, nucleotide metabolism

Address correspondence to Margaret Sällberg Chen, Margaret.Chen@ki.se.

The authors declare no conflict of interest.

See the funding table on p. 17.

Emerging research indicates that tumor-associated microbes have a significant role in cancer initiation and progression (1). Nucleotide metabolism contributes to uncontrolled tumor growth and plays a crucial role in chemotherapy (2), which counteracts their excessive consumption by tumor cells (3). A growing body of publications demonstrates that microbes are a common component of tumor microenvironment (TME) in breast, lung, ovary, pancreas, and several other types of cancer, and that they could promote tumor growth as well as therapy resistance (1). Microbes generally have extraordinary survival skills due to their natural ability to adapt in harsh environments through genetic evolution. Understanding how they enhance key pathways, metabolize environmental factors, and deploy survival mechanisms to cope with the challenging conditions of the TME is of great interest (4).

Pancreatic cancer (PC) is the third most lethal cancer globally, with a 5-year overall survival rate of only 9%–11%. Pancreatic ductal adenocarcinomas (PDAs) represent the most common histological type of PC and account for 85%–90% of all diagnosed PC cases (5). Treatment options include surgical resection and chemotherapy, depending on the stage of disease dissemination. In other gastrointestinal cancers, oncobacteria such as *Fusobacterium nucleatum* and *Helicobacter pylori* are recognized risk factors for cancer progression (6). While the pancreas was traditionally assumed sterile, accumulating studies have reported the presence of local microbiota in PC and its tumor precursors. These include species originating from the gut and oral residents, such as *Fusobacterium nucleatum, Enterobacter asburiae, Klebsiella pneumoniae, Citrobacter freundii,* and *Enterobacter cloacae* (1, 7). The presence of the tumor microbiome has been reported to associate with increased inflammation and the "basal-like" subtype of pancreatic ductal adenocarcinoma (PDAC), which is a more aggressive phenotype (1, 7, 8). In addition to the potential role of tumor bacteria in promoting neoplastic processes, microbial interference with cancer treatment is emerging as a clinical challenge (9). Gut-resident microbes are increasingly known to cause microbial-drug interference, and an important factor in altering drug bioavailability and efficacy, including chemotherapeutic drugs (10). Several publications also found that tumor-colonizing Gammaproteobacteria can metabolize gemcitabine, a frontline chemotherapy drug used in the treatment of PC (9, 11, 12). While these new insights prompt more understanding of microbial modification of cancer drugs, most studies have focused on laboratory strains with little clinicopathology details of the tumor origin. As a result, insight into the microbial adaptation to tumoral environmental context remains limited.

We recently described cultivable bacterial strains isolated from intraductal pancreatic mucinous neoplasms (IPMNs), a major type of pancreatic cystic neoplasm regarded as precursor lesions to PDAs (13). The bacterial strains were isolated from IPMN tumors in sterile conditions during pancreatic surgery. For evaluation of the malignant progression to invasive cancer, resected IPMN tumor tissues were classified by clinical pathologists to either low-grade or high-grade dysplasia (LGD or HGD) or associated with invasive cancer (IC) using established criteria for IPMNs (14). We then assessed the potential of the intratumoral microbes to interact with chemotherapy drugs through a series of functional analyses, including drug interaction assays and whole-genome sequencing (WGS). Most of the tumor-derived strains were capable of interacting with nucleoside analogs commonly used to treat pancreatic cancer and showed genomic enrichment of purine and pyrimidine metabolism pathways. These features suggest microbial adaptation to the aberrant nucleotide metabolism of pancreatic tumors (15), representing a hallmark of tumor biology and a new therapeutic target. This adaptation likely underlies the tumor bacteria's ability to degrade nucleoside and pyrimidine analogs and persist within tumor lesions.

## RESULTS

### Phenotypic characterization of IPMN bacterial strains

We determined the growth pattern of IPMN bacterial strains from the tumor lesions that were classified as either low-grade (LGD), high-grade dysplasia (H), or PDAC invasive cancer (IC). Because PDAC and invasive cancer are clinically interchangeable, they are hereafter referred to as "IC". The doubling time (DT) and growth curves of these strains are shown in Fig. 1A and B; Table S1, along with control non-pancreatic strains (Gammaproteobacteria *Escherichia coli* d12 and *E. coli* 25922). We observed that the fastest-growing strains were *Streptococcus anginosus* (C2, H2), *Enterococcus faecium* (H2), and *Klebsiella oxytoca* (H1), which all came from HGD or IC cases or the non-pancreatic strains. Meanwhile, the slowest was *Enterobacter cloacae* (L2) from an LGD case (DT at 30–34 vs 61 minutes). From these data, exponential phases of individual strains were identified for subsequent functional experiments.

### Sensitivity of IPMN-derived strains to chemotherapeutic nucleoside and pyrimidine analogs

Chemotherapeutics 5-Fluorouracil (5-FU) and gemcitabine are well-known chemo drugs for the treatment of PC. The 5-FU is an uracil analog that inhibits the activity of thymidylate synthase, while gemcitabine is a nucleoside analog that inhibits

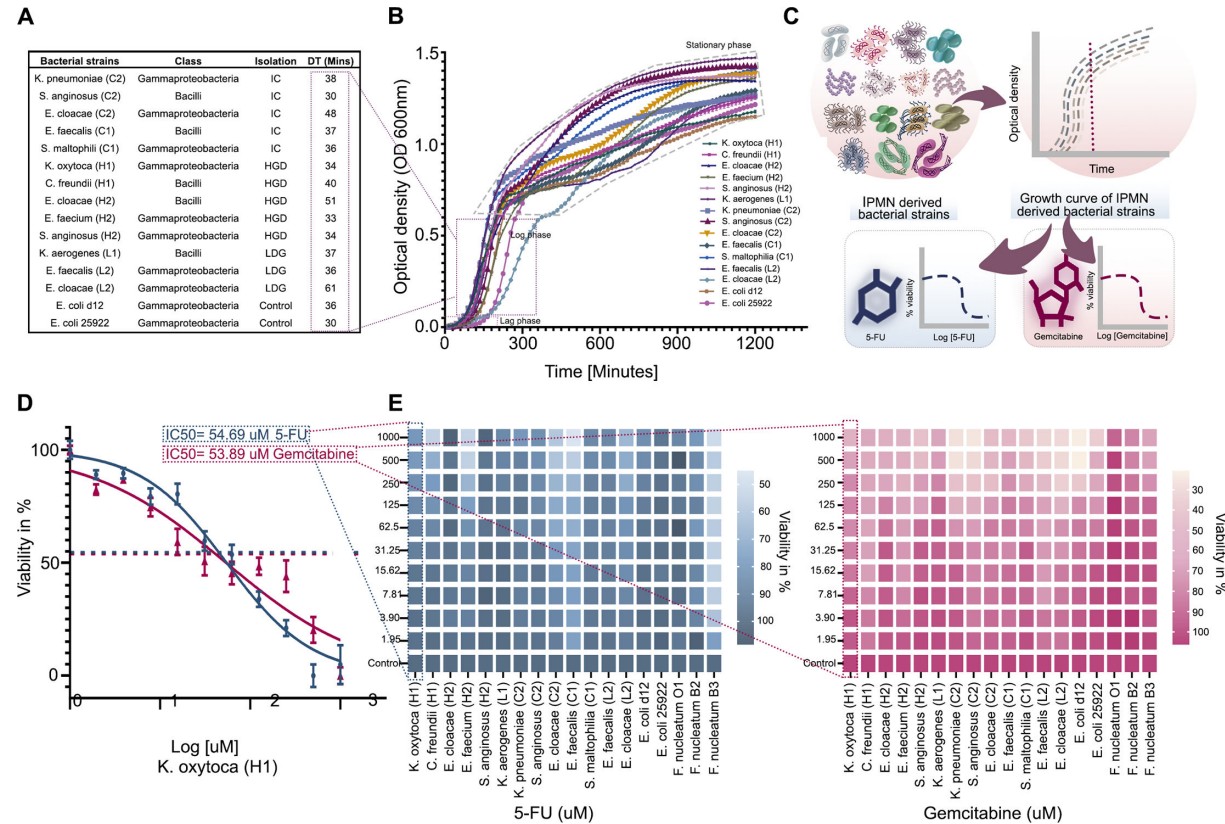

**FIG 1** Phenotypic characterization of sensitivity of IPMN-derived strains to 5-fluorouracil (5-FU) and gemcitabine. (A) IPMN-derived bacterial strains: classification, isolation source, and DT and control strains *E. coli* 25922 and *E. coli* d12. (B) Growth curves of IPMN-derived bacterial strains and the controls. The x-axis shows time (minutes), and the y-axis indicates optical density (OD) at 600 nm (mean ± SD, *n* = 3). (C) Schematic overview of the experimental design for determining IC50 values for selected IPMN-derived strains, with *E. coli* d12, *E. coli* 25922, and *F. nucleatum* strains used as controls. (D) Representative dose-response curve used to calculate IC50 for IPMN-derived *K. oxytoca* (H1) following incubation for 24 hours in brain heart infusion (BHI) media. Data were normalized, log transformed, and fitted using the Gompertz equation (mean ± SD, *n* = 3). (E) Heatmap depicting the percentage viability of individual bacterial strains in response to twofold serial dilutions of 5-FU and gemcitabine. Viability was normalized to untreated controls set at 100% (mean ± SD, *n* = 3). Left = 5-FU, right = gemcitabine.

ribonucleotide reductase (9). The potential impact of 5-FU and gemcitabine on IPMN-derived bacteria was tested through half-maximal inhibitory concentration (IC50) as shown in Fig. 1C and D. As illustrated for *K. oxytoca* (H1), the antimicrobial effect of 5-FU was 54.69 and 53.89 µM for gemcitabine, as determined by IC50 testing. Summarized in Fig. 1E; Fig. S1, and Table S2, our data indicate that while gemcitabine has an antimicrobial effect on all these pancreatic strains (IC50 at 4.6–142 µM), 54% (7 out of 13) of the IPMN-derived strains are resistant to or can endure 5-FU. The latter included *K. pneumoniae* (C2), *S. anginosus* (C2), and *Stenotrophomonas maltophilia* (C1) isolated from invasive cancer, as well as *E. cloacae* (H2) and *S. anginosus* (H2) from HGD, and *Enterococcus faecalis* (L2) and *Klebsiella aerogenes* (L1) from LGD, and the non-pancreatic bacterial strains *E. coli d12*, *E. coli 25922*. Other non-pancreatic strains, *F. nucleatum O1* and *F. nucleatum B2*, are resistant to both drugs, and *F. nucleatum B3* was sensitive to both. These results indicate that the antimicrobial effect of 5-FU and gemcitabine on IPMN tumor-derived isolates varies, with general sensitivity to the nucleoside analog gemcitabine and frequent resistance to the pyrimidine analog 5-FU.

## Metabolization of nucleoside and pyrimidine analogs by IPMN-derived bacterial strains

Next, we tested if IPMN-derived strains could metabolize these chemo drugs. Motivated by previous studies on microbiota-drug interactions and physiologically relevant range (9, 11, 12), the final drug concentrations of 4 or 40 µM were chosen in our following experiments. We incubated individual strains with or without respective drug. Thereafter, the conditioned supernatant was cleared of live bacteria through sterile filtration. As depicted in Fig. 2A, conditioned supernatants were then evaluated for their drug activity, i.e., killing of PANC-1 cancer cells known to be sensitive to these drugs. We observe that while 5-FU and gemcitabine effectively reduced the PANC-1 cell viability (Fig. 2B) as a preincubation of 5-FU (left panel) with *E. cloacae* (H2), *E. faecium* (H2), *S. anginosus* (H2), *E. faecalis* (L2), *S. maltophilia* (C1), or *F. nucleatum* O1 rendered it significantly less effective, and the effect of gemcitabine (right panel) was reduced mainly by the Gammaproteobacteria; *K. oxytoca* (H1), *Citrobacter freundii* (H1), *E. cloacae* (H2), *K. pneumoniae* (C2), and *K. aerogenes* (L1), and the *Bacillus S. anginosus* (C2). At a reduced drug concentration (4 uM), the same strains again affected the 5-FU effect on PANC-1, especially *E. cloacae* (H2), *E. faecium* (H2), *and S. anginosus* (H2). For gemcitabine at 4 uM, none of the pancreatic strains posed any effect (Fig. S2A). We then extended our analysis to include other pancreatic cancer cell types, AsPC-1 and Capan-2. The AsPC-1 cells (Fig. 2C; Fig. S2B) also showed a profound loss of sensitivity to 5-FU and gemcitabine caused by most pancreatic strains. Capan-2 cells, on the other hand, appeared less sensitive to the drugs and were, in general, less affected by bacterial-modified drugs (Fig. S2B). The cell viability was not rescued by control bacterial media generated without the drug (data not shown), suggesting that other factors besides drug degradation supporting survival capabilities were not present. A heat map summary of the IPMN strains' effect on the drug toxicity in these pancreatic cancer cell types is shown in Fig. 2D. These results indicate that IPMN-derived bacteria have an intrinsic ability to influence cancer drugs to varying degrees, rendering pancreatic cancer cells more resistant to 5-FU and gemcitabine.

## IPMN-derived Gram-negative strains are more functionally enriched compared to Gram-positive strains

Next, we sought to capture the genetic mechanism of IPMN bacterial strains to understand how they may have adapted to the IPMN environment. WGS analysis allowed us to look closer at Gammaproteobacteria (Gram-negative) and Bacilli (Gram-positive) strains for comparisons at inter- and intra-species levels. Two reference chemotherapy-resistant mutant strains of gut origin, *E. coli* MG1655 and *Bacteroides ovatis* DSM1896 (5-FU-resistant mutants 5-FU-R-M) (12), were included. WGS of all these strains (Gammaproteobacteria and Bacilli) and (5-FU-R-M) enabled a comparative analysis, in which key factors

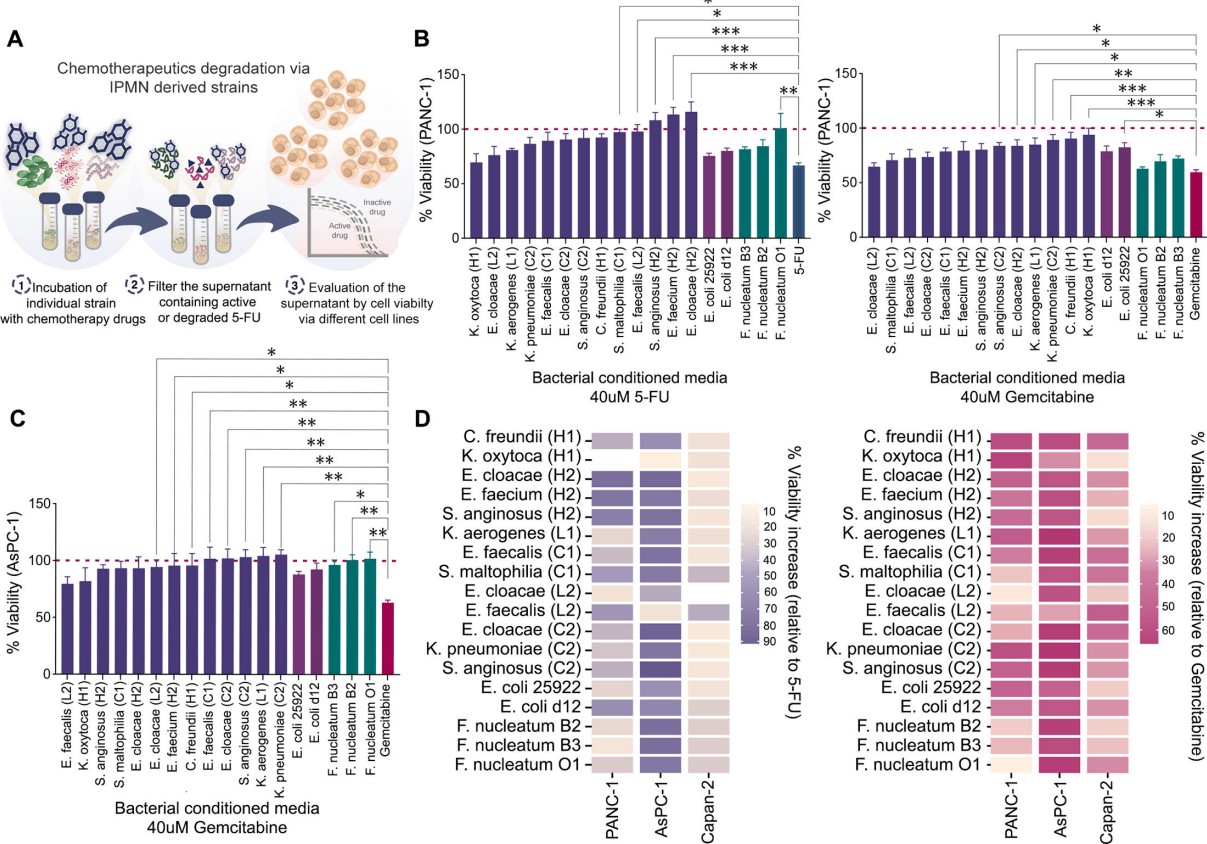

**FIG 2** IPMN-derived strains metabolize 5-FU and gemcitabine, altering drug sensitivity in pancreatic cancer cells. (A) Schematic illustration of the experimental setup for bacterial degradation of chemotherapeutic drugs. 5-FU and gemcitabine were incubated for 4 hours with or without individual bacterial strains. Following incubation, supernatants were filtered through a 0.2 µM filter. Drug activity was assessed using the CCK-8 assay, and 4 and 40 µM of the indicated drug were tested. Mock-incubated drugs served as controls for calculating relative drug activity. (B and C) Quantification of PANC-1 and AsPC-1 cell viability following exposure to bacteria-conditioned drug supernatants. Chemotherapy drug 5-FU and gemcitabine (40 µM) were first incubated with IPMN-derived strains, or *E. coli* and *F. nucleatum* for 4 hours, filtered, diluted, and added to cancer cell cultures. Cell viability was measured after 48 hours in the CCK-8 assay and expressed as percentage viability relative to mock-treated cells. Active drug controls were processed identically. Experiments were repeated twice with three independent biological replicates each. The statistical significance was examined using the Student's *t*-test. Statistical analysis was performed using Student's *t*-test and one-way analysis of variance (ANOVA). Significance levels: ns ($P > 0.05$), * ($P < 0.05$), ** ($P < 0.01$), *** ($P < 0.001$). (D) Heatmap summarizing the percentage increase in cell viability relative to active drug treatment across the PANC-1, AsPC-1, and Capan-2 cell lines. Left panel = 5-FU; right panel = gemcitabine.

of interest included pathways enriched in relation to the environment where each strain had been recovered (gut and tumor dysplasia grades). Table S3 summarizes the annotation statistics, including tRNA, rRNA CDS, hypothetical CDS, and genome quality. A circular diagram of the genomic annotations, such as GC skews, GC contents, drug targets, virulence factor genes, antimicrobial resistant genes, CDS, and other information is given in Fig. S3A. Pathway enrichment analysis conducted using rapid annotation subsystem technology (RAST) to classify superclass pathways revealed that Gammaproteobacteria of IPMN and 5-FU-R-M strains were functionally enriched by twofold compared to Bacilli strains of IPMN. This enrichment was particularly notable in pathways related to energy, membrane transport, metabolism, protein processing, regulation, and cell signaling, and stress response, defense, and virulence (Fig. 3A).

We then focused on superclass pathways of metabolism and stress response, defense, and virulence for further analysis (Fig. 3B and C). Among metabolism pathways, IPMN-derived Gammaproteobacteria strains show a similar gene count to 5-FU-R-M strains, unlike the Bacilli strains that exhibit low numbers of enriched genes in each pathway (Fig. 3B). On the stress response, defense, and virulence pathways, *E. cloacae* strains were highly enriched in genes associated with invasion and intracellular resistance compared

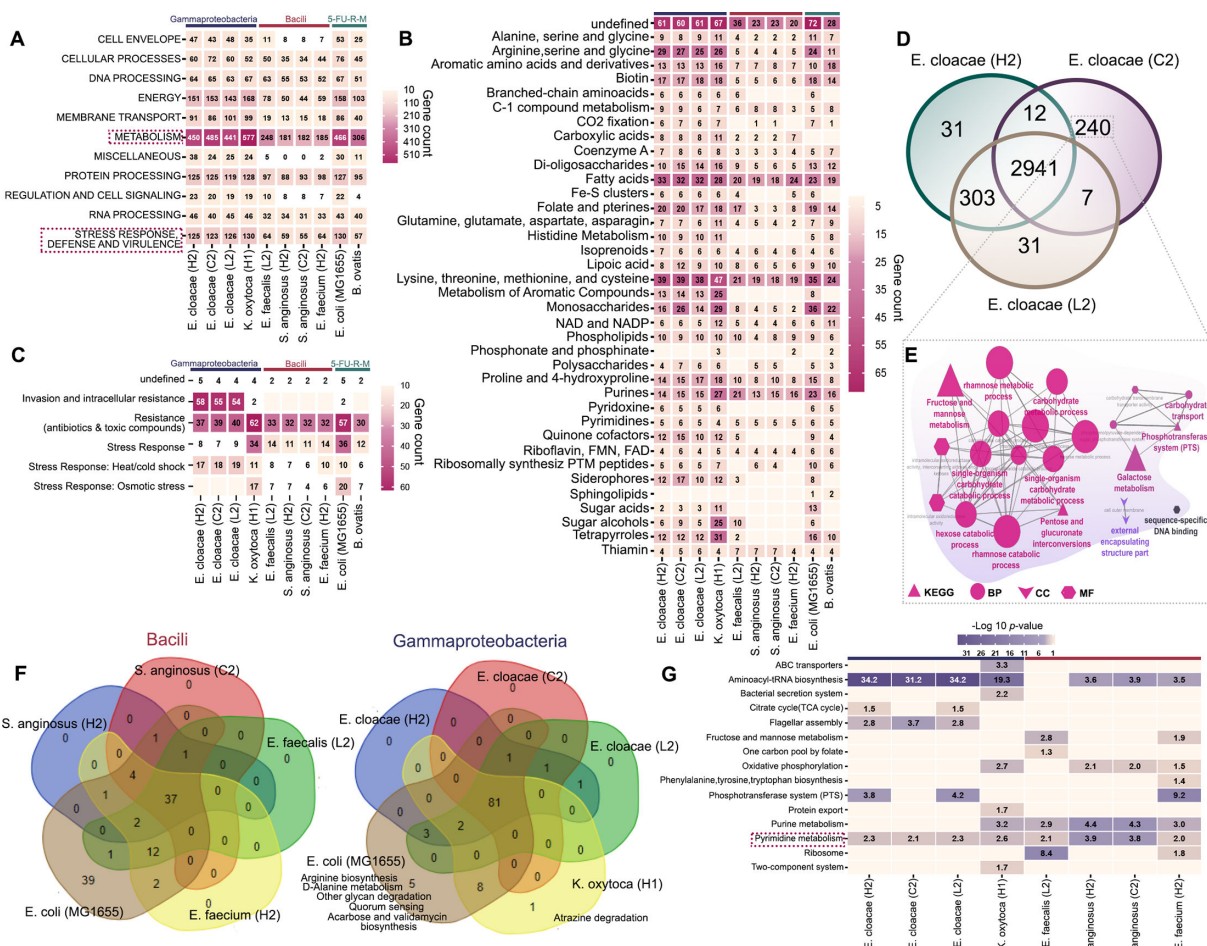

**FIG 3** IPMN-derived Gammaproteobacterial strains were functionally enriched compared to Bacilli strains. (A) Superclass pathway analysis of IPMN-derived Gammaproteobacteria, Bacilli strains, and 5-FU resistant mutant (5-FU-R-M) strains (*E. coli* MG1655 and *B. ovatis*) was analyzed using WGS, RAST, and The Seed (16, 17). The results indicated greater functional enrichment in Gammaproteobacteria compared to Bacilli. (B and C) Superclass pathways analyses—covering metabolism and stress response to virulence factors—were conducted using RAST and The Seed (16, 17). The profiles of IPMN strains were compared to 5-FU-R-M strains (*E. coli* MG1655 and *B. ovatis*), revealing consistent enrichment among Gram-negative strains. (D) Venn diagram illustrates shared and unique expressed genes in *E. cloacae* strains isolated from HGD, LGD, and IC based on evolutionary gene genealogy Non-supervised Orthologous Groups (EggNOG) analysis (18). A total of 2,941 genes were mutually expressed. Unique gene counts were *E. cloacae* (H2) (n = 31), *E. cloacae* (L2) (n = 31), and *E. cloacae* (C2). (E) Unique genes from *E. cloacae* (C2) isolated from invasive cancer were analyzed by network analysis using Cytoscape (19) and network enrichment of Gene Ontology (GO) pathways, including biological process (BP; Elipse), cellular compartment (CC; V shape), molecular functions (MF; hexagonal), and Kyoto Encyclopedia of Genes and Genomes (KEGG) pathways (triangle). Statistical significance was determined using Bonferroni step-down correction (*P* < 0.05). (F) Venn diagram shows KEGG pathways enrichment in Bacilli strains: *S. anginosus* (H2), *S. anginosus* (C2), *E. faecalis* (L2), *E. faecium* (H2), and the 5-FU resistant-mutant strain *E. coli* MG1655. Thirty-seven pathways were shared among all Bacilli, while *E. coli* MG1655 strain had 39 unique pathways (left panel). For Gram-negative strains, 81 mutual pathways, while *E. coli* MG1655 and *K. oxytoca* had 5 and 1 unique pathways, respectively. (G) KEGG pathways significantly enriched (*P* < 0.05, Bonferroni corrected) in IPMN-derived Gammaproteobacteria and Bacilli strains. Pyrimidine metabolism was notably enriched in all the Gammaproteobacteria and Bacilli strains.

to the Bacilli and 5-FU-R-M strains. Pathways related to antibiotics and toxic compound resistance and stress response were highly enriched in *K. oxytoca* (H1) at levels comparable to those seen in 5-FU-R-M *E. coli* MH1655. Stress response to heat and cold shock pathways was significantly enriched in *E. cloacae* strains, whereas the osmotic stress pathway was absent in *E. cloacae* strains but highly enriched in *K. oxytoca* (H1) and *E. coli* MG1655, as shown in (Fig. 3C). A further breakdown of superclass pathways into class pathways, highlighting the differences in pathways enrichment between Gammaproteobacteria and Bacilli strains, is provided in Fig. S3B.

We hypothesized that the tumor neoplastic grade (environment) had influenced the bacterial genomic characteristics. To test this, we used evolutionary gene genealogy Non-supervised Orthologous Groups (eggNOG) and identified all genes from orthologous groups through the WGS. Focusing on *E. cloacae* isolates from LGD, HGD, and IC, we identified 2,941 genes shared among these strains, with IC-derived *E. cloacae* (C2) carrying more unique genes than L2 and H2-derived strains (240 vs 31) (Fig. 3D). Kyoto Encyclopedia of Genes and Genomes (KEGG) and Gene Ontology (GO) biological process, cellular compartments, and molecular functions conducted ($P < 0.05$ with Bonferroni step-down correction) for *E. cloacae* (C2) indicated 240 unique genes, which are highly associated with fructose and mannose metabolism, single organism carbohydrate and catabolic process, galactose metabolism, phosphotransferase system, pentose and glucuronate interconversions, and other pathways as illustrated in Fig. 3E. These pathways are primarily involved in glucose metabolism, potentially reflecting the environmental influence from the pancreatic tumor lesion (20). A potential adaptation to the upregulation in glycolysis metabolism in tumor cells, known as the Warburg effect, e.g., rapid consumption of glucose in tumor cells (21). In conclusion, our result indicates that IC-derived *E. cloacae* (C2) overexpresses genes composing unique enriched pathways of glucose consumption not found in HGD- and LGD-derived *E. cloacae* isolates.

Next, we applied KEGG pathway analysis to gain a more comprehensive understanding of the differences between the Bacilli and Gammaproteobacterial strains, by using *E. coli* MG1655 as a reference strain. Among Bacilli strains, we identified 37 pathways shared with *E. coli* MG1655, with 39 unique to *E. coli* MG1655 (Fig. 3F, left). Among Gammaproteobacteria strains, 81 pathways were shared with *E. coli* MG1655, with five unique to *E. coli* MG1655 and one unique to *K. oxytoca,* as shown in Fig. 3F, right. We next compiled all significantly enriched KEGG pathways ($P < 0.05$ with Bonferroni step-down correction) and found pyrimidine metabolism pathways present in all IPMN-derived strains (Fig. 3G), suggesting that pyrimidine metabolism is significantly enriched across IPMN-derived strains. Previous studies have shown that fluoropyrimidines disrupt pyrimidine metabolism in both *E. coli* and mammalian cells (12). Nucleotide metabolism is a critical pathway consisting of purine and pyrimidine metabolism, while purine and pyrimidine molecules are building blocks for DNA replication and are essential for cell proliferation (2). Our results suggest the possibility that IPMN-derived strains retain enriched nucleotide metabolism in their genomes, thus enabling metabolization of nucleoside/pyrimidine analog gemcitabine and 5-FU.

## Comprehensive transcriptomic analysis of *E. coli* capable of degrading 5-FU

To verify the hypothesis above, we included transcriptomic data from publicly available data sets to study bacterial interactions with 5-FU. An earlier study (12) defined bacterial metabolic pathway usage in 5-FU exposure conditions by conducting transcriptional profiling. In that experiment, *E. coli* MG1655 was treated with 5-FU at sub-MIC dosage against the vehicle control strain. We identified a transcriptomic change that involved 432 upregulated and 600 downregulated genes in the aerobic conditions (Fig. 4A). The upregulated KEGG pathways were pyrimidine metabolism, homologous recombination, and thiamine metabolism, while downregulated pathways encompassed glycan degradation, bacterial chemotaxis, ABC transporters, butanoate metabolism, tricarboxylic acid (TCA) cycle, among others (Fig. 4B). For pathway enrichment analysis, we conducted networking analysis using Cytoscape, which identified significant GO pathways of biological process that were enriched in nucleotide metabolism, containing pyrimidine-containing compound metabolic pathways, pyrimidine metabolism, purine metabolism, thiamine diphosphate metabolic process, organonitrogen compound biosynthetic process, and other pathways ($P < 0.05$ Bonferroni step-down correction) (Fig. 4C) while downregulated pathways are shown in Fig. 4D. Similarly, the anaerobic condition with 5-FU treatment at sub-MIC is shown in Fig. 4E, where our analysis identified upregulated KEGG pathways that included the pyrimidine metabolism,

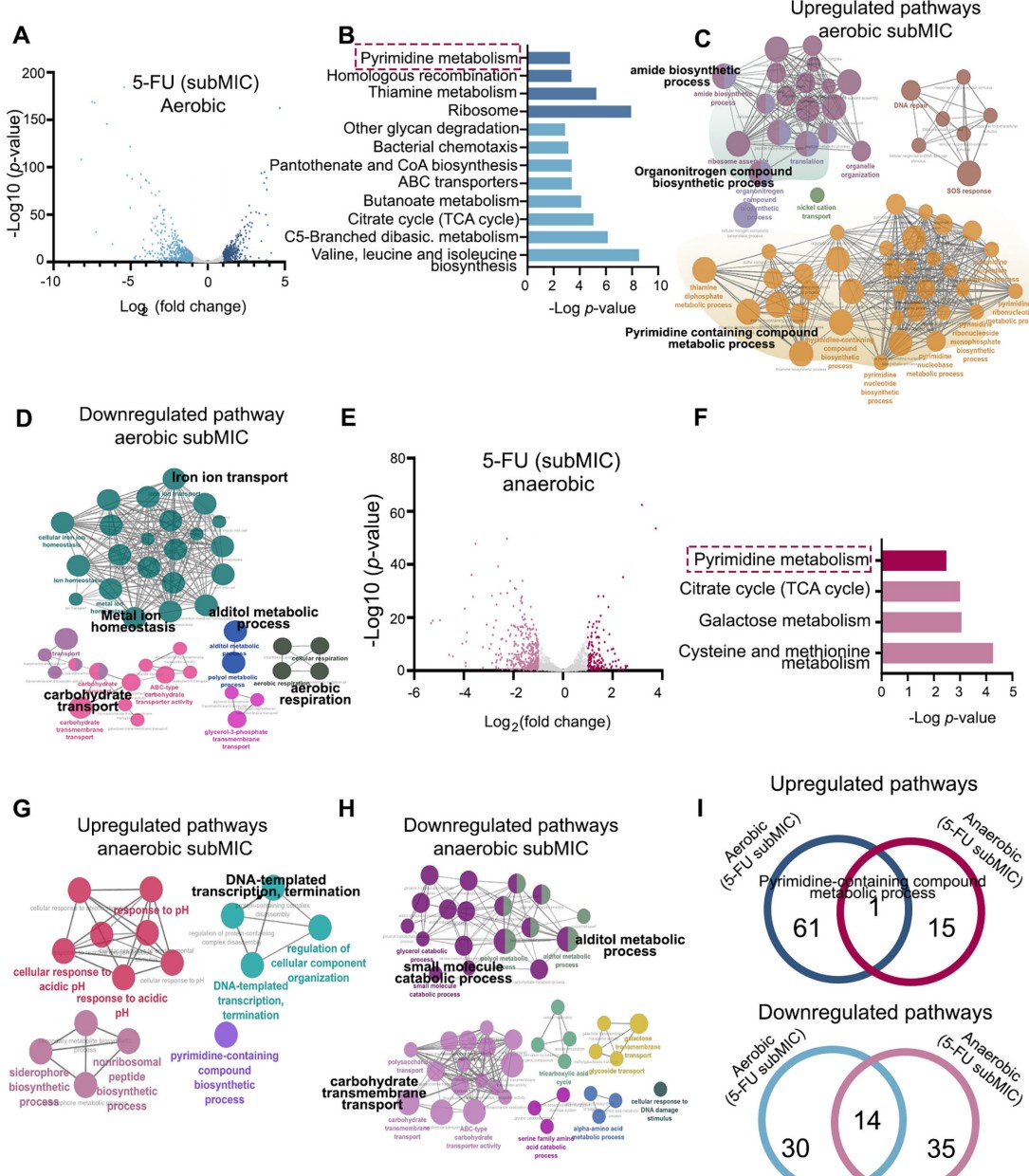

**FIG 4** Transcriptomic analysis of 5-FU induced response and survival mechanism of *E. coli* MG1655. (A) Volcano plot showing differentially expressed transcripts of *E. coli* MG1655 exposed to 5-FU under aerobic conditions (data from reference 12). Upregulated genes are shown in dark blue and downregulated genes in light blue color with (FDR < 0.1, |log2 fold-change| > 1). (B) KEGG pathway enrichment analysis of differentially expressed genes (DEGs) following 5-FU treatment under aerobic conditions. Pathways were identified using Cytoscape, with significance determined by Bonferroni step-down correction ($P < 0.05$). Upregulated pathways are shown in dark blue (top), and downregulated pathways in light blue (bottom). (C and D) GO enrichment analysis for biological processes associated with upregulated and downregulated DEGs under aerobic conditions. Analyses were performed in Cytoscape with Bonferroni step-down correction ($P < 0.05$). (E) Volcano plot of differentially expressed transcripts in *E. coli* MG1655 exposed to 5-FU under anaerobic conditions. Upregulated genes are shown in red, and downregulated genes in pink (FDR < 0.1, |log2 fold-change| > 1). (F) KEGG pathway enrichment analysis of DEG following 5-FU treatment under anaerobic conditions. Upregulated pathways (top, pink) and downregulated pathways (bottom, red) ($P < 0.05$, Bonferroni step-down correction). (G and H) GO enrichment analysis of biological processes for upregulated and downregulated DEGs under anaerobic conditions, using Cytoscape with Bonferroni step-down correction ($P < 0.05$). (I) Venn diagram showing shared GO biological process pathways regulated by 5-FU under aerobic and anaerobic conditions. Upregulated pathways are shown at the top and downregulated pathways at the bottom ($P < 0.05$, Bonferroni step-down correction).

whereas downregulated pathways were enriched in TCA cycle, galactose metabolism, and other pathways, as shown in Fig. 4F. The biological process for anaerobic conditions is shown in Fig. 4G and H. Our analysis could verify the mutual pathways enriched in 5-FU treated *E. coli* MG1655 in both aerobic and anaerobic conditions, that pyrimidine-containing compound metabolic process was upregulated in both conditions (Fig. 4I, top). For downregulated genes (Fig. 4I, bottom), 14 pathways were shared in both conditions, while 30 were unique to aerobic conditions and 35 to anaerobic conditions (Fig. 4I, bottom).

## IPMN-derived Gammaproteobacteria nucleotide metabolism links to the IPMN neoplastic grade

To gain further insight into the IPMN-derived strains, we analyzed specialty genes from their WGS using the Pathosystems Resource Integration Center (PATRIC) (22). This included antibiotic resistance genes (mapped with CARD [23]), drug targets (mapped with Drugbank and TTD [24]), virulence factors (mapped with VFDB [25] and Victors [26]), and transporters (mapped with TCDB [27]). Based on this, we assessed functionally enriched pathways using KEGG and Gene Ontology for biological process, cellular component, and molecular functions via Cytoscape ($P < 0.05$ with Bonferroni step-down correction). We observed that *E. cloacae* (H2, L2, and C2) strains were significantly enriched in nucleotide metabolism, consisting of pyrimidine-containing compound metabolism, pyrimidine metabolism, purine metabolism, and other pathways, as illustrated in Fig. 5A; Fig. S4A. In *E. coli* MG1655, nucleotide metabolism, pyrimidine-containing compound metabolism, pyrimidine metabolism, purine metabolism, organonitrogen biosynthetic process, and small molecule biosynthetic process and other pathways were also enriched ($P < 0.05$ with Bonferroni step-down correction) shown in Fig. 5B. The enriched pathways for our other strains ($P < 0.05$ with Bonferroni step-down correction) are summarized in Fig. S4B through F. For specialty genes, significant KEGG pathways ($P < 0.05$ with Bonferroni step-down correction) are shown in Fig. 5C. Most enriched pathways were found in Gammaproteobacteria, while purine metabolism and RNA degradation are exclusively enriched in Bacilli strains. This indicates that Gammaproteobacteria exhibit greater functional enrichment compared to Bacilli strains in terms of specialty genes.

Next, we examined the KEGG pathways related to nucleotide metabolism, which includes both purine and pyrimidine metabolism, and specifically focused on pyrimidine metabolism to meet the relevance of 5-FU as a pyrimidine analog. Bacterial strains employ various mechanisms and pathways in pyrimidine metabolism (29). We analyzed the entire pyrimidine metabolism pathway by their gene count for each enzyme involved in selected Gammaproteobacteria, Bacilli, and 5-FU-resistant-mutant (5-FU-R-M) strains (Fig. 5D). Our first notion was that the expression (gene count) of cytosine deaminase (CDD) EC 3.5.4.5, responsible for gemcitabine degradation, was doubled in 5-FU-R-M strains (9). Moreover, Pre A and Pre T (EC 1.3.1.1) dihydropyrimidine dehydrogenase (NAD+), which is known to inactivate 5-FU in *E. coli* strains, was found only in *E. coli* MG1655 (12). We also considered the Rut pathway (pyrimidine degradation), which consists of seven proteins required for *E. coli* K-12 to utilize uracil as a source of nitrogen (30). We found that proteins related to the Rut pathway, including pyrimidine oxygenase (Rut A, EC 1.14.99.46) and peroxyuridoacrylase (Rut B to Rut G, EC 3.5.1.110), were present in Gammaproteobacteria strains and absent in Bacilli strains. This suggests that Gammaproteobacteria and Bacilli strains employ different mechanisms for pyrimidine metabolism, as the Rut pathway was not identified in Bacilli strains. A heat map illustrating the enzymes, their EC number, and gene count is shown in Fig. 5D.

We next revisited our earlier drug degradation results, noting that *E. cloacae* (L2) showed the least degradation of 5-FU and gemcitabine (Fig. 2D). We compared its genome to those of chemotherapy drug degraders *E. faecium* (H2), *E. cloacae* (H2), *S. anginosus* (H2), and *K. oxytoca* (H1), as illustrated in a Venn diagram. We found that *E. cloacae* (L2) differed from the other strains by 28 unique genes, of which no significant

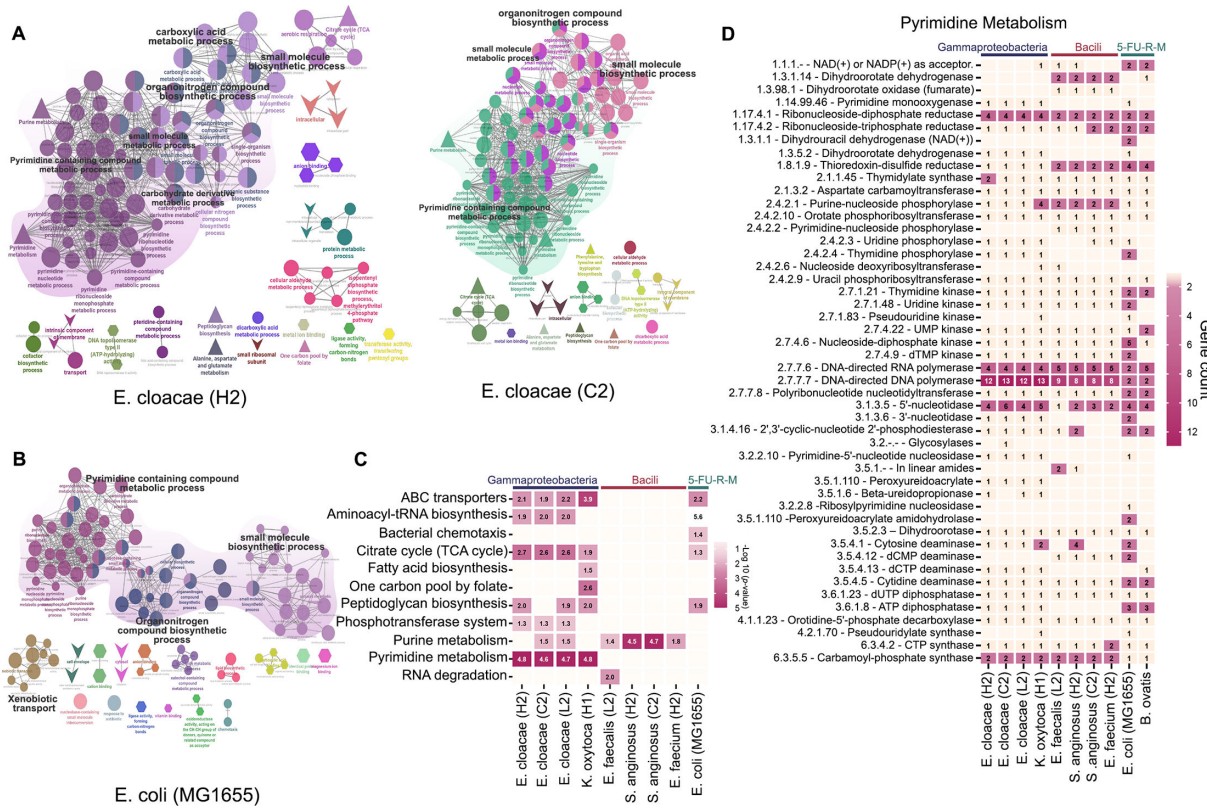

**FIG 5** Functional pathway analysis reveals enrichment of pyrimidine metabolism in IPMN-derived strains as a survival mechanism. (A) GO terms for biological processes, molecular functions, cellular components, and KEGG pathways were visualized based on the presence of specialty genes, including antibiotic resistance genes (mapped using CARD [23]), drug targets (Drugbank and TTD [24]), virulence factors (VFDB [25] and Victors [26]), and transporters (TCDB) (27). (B) Enrichment analysis was performed with Bonferroni step-down correction ($P < 0.05$). Left panel: *E. cloacae* (H2); right panel: *E. cloacae* (C2). (C) KEGG pathway enrichment (28) across all the IPMN-derived Gammaproteobacteria and Bacilli strains ($P < 0.05$ Bonferroni step-down correction). (D) Pyrimidine metabolism pathways were reconstructed using enzyme commission (EC) numbers and gene counts for IPMN-derived Gammaproteobacteria, Bacilli, and 5-FU-resistant mutant strains (*E. coli* MG1655 and *B. ovatis*) using the PATRIC BVBRC database.

pathways could be identified (Fig. S5A), whereas significant pathways emerged from unique genes of *K. oxytoca* (H1), *S. anginosus* (H2), and *E. faecium* (H2) (Fig. S5B through D). This result, combined with the slow phenotypic growth in *E. cloacae* (L2) (Fig. 1B), could have contributed to the low ability to metabolize drugs shown by the strain.

## Common pathways in bacteria isolated from different neoplastic environments

Microbial strains can adapt to their environment by altering their genetic makeup and community structure for improved survival (31). To test whether bacteria isolated from different neoplastic environments share genetic similarities, microbial genes were extracted using eggNOG. Shared genes between IPMN grades are illustrated in Fig. 6A and B. In LGD strains, 682 genes were shared, while 775 in *E. faecalis* (L2) and 2,600 in *E. cloacae* (L2) were unique (Fig. 6A). For HGD strains, 490 genes were shared between all four strains (Fig. 6B). This indicates that Gammaproteobacteria and Bacilli share similar genes, which may constitute the survival mechanism in an IPMN environment. For LGD strains, the enriched pathways included small molecule metabolic process, pyrimidine, DNA repair and homologous recombination, RNA processing, glycosamino-glycan metabolic process ($P < 0.05$ with Bonferroni step-down correction) (Fig. 6C). While for HGD, pyrimidine metabolic pathways, small molecule metabolic process, carboxylic acid metabolic process, cell wall organization, and others were overrepresented ($P < 0.05$ with Bonferroni step-down correction) (Fig. 6D). In IC strains, 517 genes were

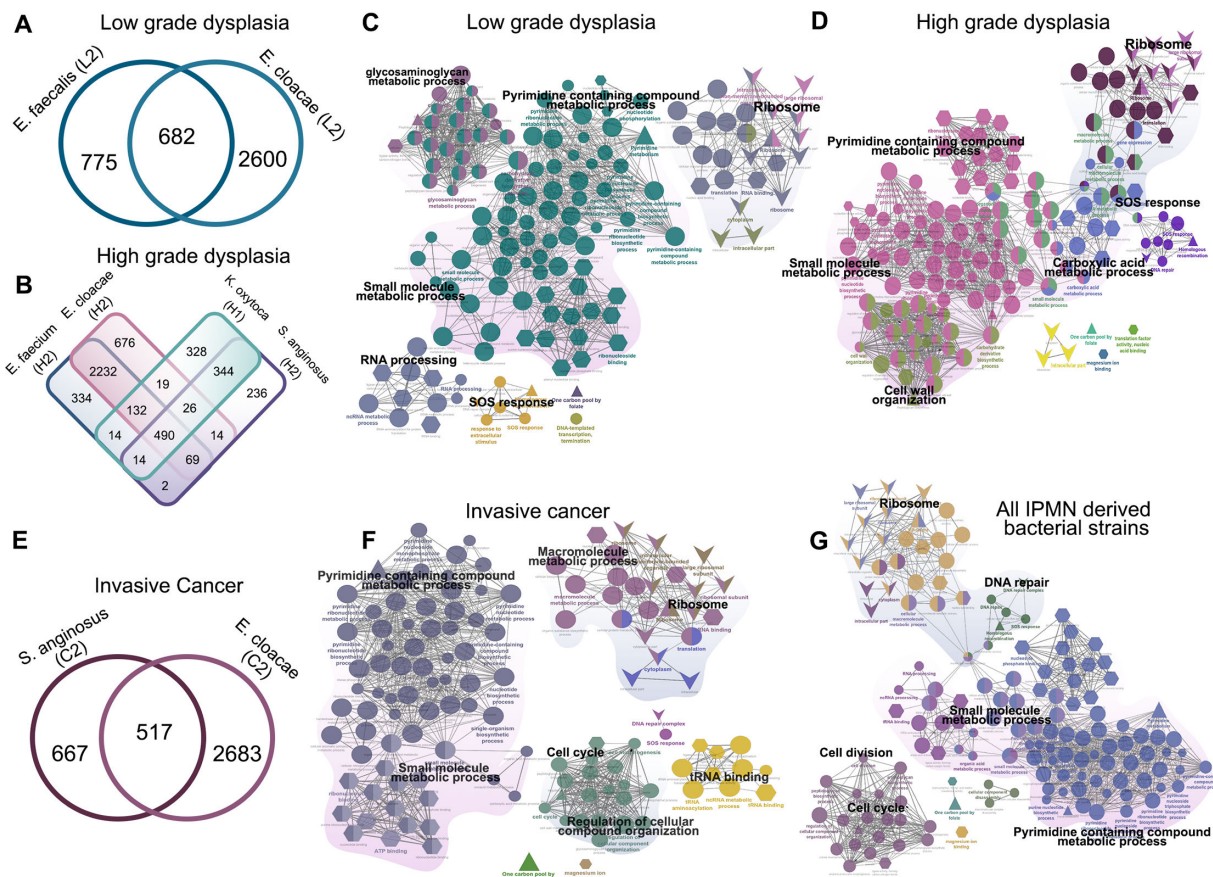

**FIG 6** Functional enriched pathways in LGD, HGD, and invasive cancer-derived bacterial strains. (A) Venn diagram comparing LGD-derived *E. faecalis* (L2) and *E. cloacae* (L2), showing 682 shared genes, with 775 and 2,600 genes unique to each, respectively. (B) Venn diagram of HGD-derived strains (*E. faecium* H2, *E. cloacae* H2, *K. oxytoca* H1, and *S. anginosus* H2) showing 490 shared genes across all strains, despite differences in Gram status. (C) Functional enrichment of the 682 shared genes from LGD-derived strains revealed significant enrichment in pyrimidine metabolism, small molecule metabolic processes, and other pathways (*P* < 0.05, Bonferroni step-down correction). (D) Similarly, 490 shared genes from HGD strains were enriched in pyrimidine metabolism, small molecule metabolism, and related processes (*P* < 0.05, Bonferroni step-down correction). (E and F) Venn diagram of invasive cancer-derived strains showed 571 shared genes, functional enrichment of those genes showed enrichment in pyrimidine metabolism, small molecule metabolic processes, and additional pathways (*P* < 0.05, Bonferroni step-down correction). (G) Across all IPMN-derived Gammaproteobacteria and Bacilli strains, 421 genes were found to be commonly shared. Functional enrichment indicated primary enrichment in pyrimidine metabolism, small molecule metabolism, DNA repair, and related pathways (*P* < 0.05, Bonferroni step-down correction).

shared between *S. anginosus* (C2) and *E. cloacae* (C2), which were enriched in pathways of pyrimidine metabolism, small molecule metabolic process, RNA binding, cell cycle, and others (*P* < 0.05 with Bonferroni step-down correction) (Fig. 6E and F). Next, we selected the shared 421 genes by the IPMN strains for a pathway enrichment analysis, which identified significant enrichment in pathways related to ribosomes, DNA repair, pyrimidine metabolism, small molecule metabolic process, cell cycle, and other pathways (*P* < 0.05 with Bonferroni step-down correction) (Fig. 6G). In conclusion, our results illustrate how these bacterial strains may have adapted to different environmental conditions at the genomic level, promoting their survival and persistence within neoplastic lesions in the pancreas.

## DISCUSSION

The TME of pancreatic cancer comprises non-malignant cells, including fibroblasts and immune cells, as well as severely hypoxic conditions (32). Within this environment, tumor-associated microbes are increasingly recognized as integral components,

potentially contributing to cancer initiation and progression. Pancreatic cancer can contain bacteria that can potentially modulate tumor sensitivity to gemcitabine (9). Members of Gammaproteobacteria have been found to metabolize gemcitabine into inactive forms through bacterial enzyme cytidine deaminase CDD (9), while the preTA operon is necessary for 5-FU inactivation by *E. coli* (11, 12). Microbes can adapt to their environments through mechanisms of drug resistance (33), horizontal gene transfer (34), plasmids and phages (35), efflux pumps (36), and microbial evolution to survive specific environmental conditions (37). Our current study here is the first to demonstrate that IPMNs, which are precursors associated with PC, also harbor tumor-associated microbes with unique survival mechanisms. Our data reveal how these microbes have adapted to the neoplastic environment of the pancreatic tumors. This finding has important clinical implications, as their adaptation and persistence may contribute to the IPMN progression and subsequent cancer therapy resistance by actively interfering with chemotherapy treatment.

Our study is a follow-up of our previous study, where culturable pancreatic micro-biota was identified in the cyst fluid of pancreatic cystic tumors in patients undergoing pancreatic surgery (7, 13). None of the patients had received neoadjuvant with either gemcitabine or 5-FU before the surgery; it is therefore noteworthy to observe the substantial drug tolerance in these tumor bacteria, particularly to the 5-FU. Most notably, nearly all these strains demonstrated ability to degrade chemotherapeutic drugs which altered the drug cytotoxic effects on pancreatic cancer cells. Consistent with previous studies reporting that various *E. coli* strains are enriched in pyrimidine metabolism and degrade 5-FU (12), our WGS analyses revealed functional enrichment of nucleotide metabolism pathways in both Gammaproteobacteria and Bacilli strains, conveyed by their specialized genes.

Survival strategies consist of bacterial drug efflux, drug transformation by metabo-lizing, and by modifying drug targets at the single-organism level (38). The microbial phenotypes, therefore, play a key role in bacterial evolution and adaptation. In our study, the slow-growing *E. cloacae* (L2) strain, which had the longest DT, may explain its limited drug degradation capacity. This aligns with previous findings showing that faster-grow-ing bacteria are generally more efficient at metabolizing drugs (39). Moreover, the IC-derived *E. cloacae* (C2) strain also had enriched pathways related to glucose metab-olism, which may be due to the Warburg effect (20, 40). Despite extensive usage of pyrimidine derivatives in anticancer drugs, emerging research indicates that bacterial adaptation remains one step ahead and hinders chemotherapy drug metabolism (12). Our findings suggest that pathways of nucleotide metabolism, such as pyrimidine and purine metabolism, were intrinsically enriched in IPMN-derived strains, likely due to their survival mechanisms. In these Gammaproteobacteria, the pyrimidine metabolism pathways were particularly enriched for the genes responsible for virulence factors, antimicrobial resistance (AMR), drug targets, and drug transporters. Pyrimidines are a rich source of nitrogen and essential biomolecules, and structural components for nucleoti-des, nucleic acids, and necessary for fulfilling crucial roles in the cell (41), and represent one of the key pathways reflecting a survival strategy in a competitive, resource-variable niche like the TME, including IPMN nucleotide-rich environments with high rates of cell turnover, necrosis, and DNA/RNA release (41).

To our knowledge, this is the first data to demonstrate that microbes from a tumor environment carry features suited to persist in TME, which in turn could serve in microbial-drug interactions, rendering them capable of metabolizing cancer drugs of nucleoside/nucleotide analogs. The latter result is also supported by three independent cancer cell types: PANC-1 (primary tumor) (42), AsPC-1 (from ascites) (43), and Capan-2 (primary tumor) (44), unlike other reports that rely on single cell type, e.g., [AsPC1(9), RKO(16), or HCT-116(12)]. Whether IPMN-associated microbes have been directly influenced by their tumor environments and therefore adapted to these conditions to survive remains to be investigated. Genome comparisons of the same bacterial strains from tumor and non-tumor origins could provide further insight. Moreover,

our Gram-negative strains also belong to the Gammaproteobacteria phylum, known to include clinically significant pathogens. These genomes are more extensively studied than those of Gram-positive strains, possibly contributing to the greater functional annotation and pathway-level insights currently available for Gram-negative bacteria. In this context, our tumor-associated bacterial strains appear to exhibit similar characteristics. Other typical features of Gram-negative bacteria, e.g., greater phenotypic and genotypic diversity, and more complex cell wall structures, also likely contribute to broader functional capabilities noted here. Another point of interest in our study is the example of *E. cloacae* strains isolated from tumors of different malignancy grades (low-grade dysplasia, high-grade dysplasia, and invasive cancer). Despite being the same species, these strains exhibited distinct gene expression profiles across different pathways, not previously reported in the context of tumor bacteria research. Additional work should also explore the differences between tumor and non-tumor origins further.

Our study has several limitations. First, our microbial-drug interaction study is limited to only two common PC drugs, and a short co-incubation time is used to examine the drug interactions. More extensive studies are needed, including quantitative and qualitative measurements on the inactivated drug compounds through, for instance, high-performance liquid chromatography. Additionally, the number of IPMN-derived strains could also have been greater and compared to bacteria isolates from patients treated with chemotherapy drugs. The WGS-related data are based on functional predictions on genomic data, and while this requires extensive and multi-level computational analyses, additional experimental validations would be needed. This could involve, for instance, genetically modified mutant bacteria.

In conclusion, our study provides new insights that extend previous research, which has primarily focused on intestinal microbes. Here, we present evidence of bacteria cultivated directly from precursor lesions associated with pancreatic cancer. We examined these microbes from the bacterial perspective, considering potential countermeasures and selection pressures imposed by the tumor microenvironment. Our findings offer mechanistic insights into the metabolic pathways utilized by tumor-associated microbes derived from IPMN precursors linked to PC. In the broader context of the nucleotide dependency of cancer development (45), our results are highly relevant, not only in terms of microbial adaptation to the tumor microenvironment but also in regard to drug metabolism, chemoresistance, microbial fitness, and niche specialization.

## MATERIALS AND METHODS

### Study population and sample collection

Cyst fluid samples were collected from patients undergoing elective surgery for suspected malignancy due to a pancreatic cystic lesion. Patients aged 18 years or older, with written informed consent and a histologically confirmed pancreatic cystic lesion, were included. All patients with a concomitant cancer who had received neo-adjuvant chemotherapy for pancreatic cancer or any previous chemotherapy for other indications were excluded. Clinical data were extracted from the electronic medical records system. The study was conducted with the permission of the Regional Ethical Review Board in Stockholm (Dnr 2015/1580-31/1) and follows the Helsinki Convention and good clinical practice. Details of the study population and sample collection are mentioned in reference 13.

### Collection of cyst fluid and histopathological diagnosis

The process of obtaining and managing the cyst fluid collections is mentioned in our previous study (13). The resected specimens were classified and graded after histopathological analysis by a specialist as LGD, HGD, and IC. In cases of moderate-grade dysplasia, they were classified as LGD according to the current WHO classification.

## Bacterial cultivation and MALDI-TOF MS analysis

Pancreas IPMN-derived bacterial strains were cultured as previously described (13). Briefly, pure colonies were obtained on agar plates and subjected to species identification using matrix-assisted laser desorption/ionization time-of-flight mass spectrometry (MALDI-TOF MS). MALDI-TOF MS scores ≥1.70 and ≥2.00 were accepted as successful confirmation at genus and species levels, respectively. Microbial cultivation was performed on blood agar and incubated under anaerobic conditions at 37°C for 48 hours. Anaerobic bacteria cultivation was performed in a strict anaerobic environment. Single colonies of identity-confirmed bacterial isolate strains were selected and cultured in 5 mL of liquid brain heart infusion (BHI) medium for 24 hours at 37°C, 200 rpm orbital shaking.

## Bioscreening of IPMN-derived bacterial strains

All experiments were carried out using original glycerol stocks. Bacterial strains were cultured according to the bacterial cultivation method given above. Optical density (OD) of suspension was measured at 600 nm (CLARIOstar multi-mode plate reader) using absorbance. Thereafter, the bacterial cultures were further diluted to an OD of 0.1. For bioscreening experiments, 10 uL of suspension at OD 0.1 mixed with 190 uL BHI media was added into honeycomb 100-well microplates #95025BIO (Bioscreen) and incubated at 37°C in Bioscreen C apparatus. Bacterial growth curves were determined by monitoring OD absorbance (600 nm) every 15 minutes to calculate the log, exponential, and stationary phases, respectively, of each bacterial strain. Bacterial doubling times were quantified at their exponential phase according to earlier descriptions (46). Three replicates per condition were used in each assay.

## Measurement of bacterial dose responses and IC50 to cancer drugs

Chemotherapy drugs 5-FU (Cat. No. HY-90006 MedChemExpress MCE) and gemcitabine (Cat. No. HY-17026 MedChemExpress MCE) were serially diluted in BHI media and plated in a 96-well plate at 2× concentration in 100 uL per well with indicated bacterial strain (exponential phase) according to protocol by Barnes et al. (47). The 96-well plates containing bacterial suspension with serially diluted drugs were covered with parafilm to prevent evaporation and incubated at 37°C for 24 hours. The next day, bacterial growth was measured using a CLARIOstar multi-mode plate reader at 600 nm. Absorbance measurements in OD were log-transformed, normalized, and IC50 values for individual bacterial strains for the calculation of dose-response curves in GraphPad Prism 10 (version 10.1.2, San Diego, California, USA) (48). Strains with an IC50 that fell within the tested concentration range were categorized as sensitive, while those with an IC50 exceeding the highest concentration tested were classified as resistant.

## Monitoring bacterial degradation of chemotherapy drugs

Chemotherapy drugs 5-FU (Cat. No. HY-90006 MedChemExpress MCE) and gemcitabine (Cat. No. HY-17026 MedChemExpress MCE), dissolved according to the manufacturer's guidance in BHI medium, and added to the indicated bacterial strains to achieve the final concentration of 4 or 40 µM. Mock controls without bacteria were used in each experiment. Suspensions were incubated for 4 hours at 37°C in a shaking incubator at 200 rpm; thereafter, they were filtered with a 0.2 µm filter syringe (Cytiva Acrodisc Syringe Filters with Supor Membrane, Sterile 17194361) to remove bacteria and cell debris. Obtained filtered conditioned media were tested on pancreatic cancer cell lines to measure the drug activity. Pancreatic cancer cell lines PANC-1 (ATCC CRL-1469), AsPC-1 (ATCC CRL-1682), and Capan-2 (ATCC HTB-80) were obtained from ATCC (Manassas, VA, USA). Each cell line was maintained in specified media according to the supplier's instructions. Cells were seeded at a density of 2,000 cells per well (100 µL) in flat-bottom 96-well plates and incubated for 24 hours; thereafter, the culture media were removed

and replaced by the conditioned drug media of twofold serial dilutions to allow another 48 hours of incubation. Cell viability was compared against mock conditioned media. The Cell Counting Kit-8 (CCK-8) (Cat. No. HY-K030, MedChemExpress MCE) was used to measure cell viability according to the manufacturer's instructions. The OD absorbance was measured at 450 nm in the (CLARIOstar multi-mode) plate reader. The experiment was repeated twice in triplicate, and the cell morphology was monitored using the Nikon TMS inverted microscope (49, 50).

## DNA extraction for whole-genome sequencing

Single bacterial colonies of MALDI-TOF MS identity-confirmed isolates were selected and cultured in 5 mL of liquid BHI medium for 24 hours at 37°C in a shaking incubator at 200 rpm. Growth was controlled with Densicheck (Densichek Plus Standards Kit-OT-21255), and $6 \times 10^8$ bacterial cells were pelleted for genomic DNA extraction by the EZ1 instrument (EZ1 Advanced XL, Cat. No./ID 9001875). EZ1 DNA Tissue Kit was used, and purified bacterial DNA was eluted in RNase-free water and stored at −20°C; thereafter, DNA libraries were made using the Illumina bead-linked transposome complex technology and sequenced at the Karolinska Institutet NGI SciLife Laboratory (51).

## Bacterial genome sequence assembly and mapping

Raw data were trimmed with Cutadapt and quality-checked with FastQC. Quality control metrics indicated high data quality, with no duplicated reads (0%), average per-sequence quality scores of 34.5 (corresponding to a base calling accuracy of 99.95%), minimal adapter contamination (<0.1%), and less than 1% overrepresented sequences. FASTQ utilities were performed using PATRIC to remove the adapters, and trimming was performed. FASTQC was used for quality checks on the raw sequence data, while Bowtie2 was used for genome alignment, and the quality of alignment was examined. The pair filter FASTQ utility was used for downstream analysis to ensure that all paired-end reads have matched reads (52). For further downstream analysis, paired trimmed FASTQ files were used.

## Comprehensive genome analysis

For comprehensive genomic analysis of each strain, paired trimmed FASTQ files were used. Bacterial comprehensive analysis included assembly, annotation, a comprehensive subsystem summary, and functional annotation of individual strains. After quality control and trimming of adapters, the resulting sequence reads were assembled into contigs using Unicycler (53). By using comprehensive genome analysis, reads for contigs, GC content, plasmids, genome length, chromosomes, and L50 were quantified. From genome annotation taxonomy, CDS, tRNA, rRNA, misc RNA, protein features, and a circular view of the genome were annotated. Genomes were annotated against taxon IDs 2027919, 571, 1351, 1328, and 1352. In the genomic circular view, specialty genes, drug targets, transporters, and virulence factors were annotated. Using RAST, microbial annotations were built by identifying protein-encoded genes and RNA. Comprehensive genomes of bacterial strains were identified using PATRIC (22). For predicting genomic function and pathways of all the selected strains, the SEED platform was used, as RAST and SEED are interconnected annotation pipelines (16, 54). For genomic annotation of each strain, functional annotation, eggNOG mapper was used, which is a bioinformatics resource providing ortholog data and comprehensive functional information for organisms across all domains (55). The genes of each organism were identified, and functional enrichment analysis was performed using GO pathways as biological process (BP), cellular compartments (CC), molecular functions (MF), and KEGG pathways were plotted with statistical significance set at $P < 0.05$ with Bonferroni step-down correction. 5-FU resistant mutant strains *E. coli* MG1655 and *B. ovatis* DSM1896 WGS data were obtained from the public database NCBI BioProjects PRJNA576932 (12).

## Transcriptomic analysis

For transcriptomic analysis, the data set from NCBI BioProjects PRJNA576932 for *E. coli* MG1655, aerobic and anaerobic culture conditions with 5-FU drug concentration at 0.5× MIC as described in Spanogiannopoulos et al. (12) was selected.

## Data visualization

IC50 data were calculated using GraphPad Prism 10 (version 10.1.2; San Diego, California, USA). WGS results were visualized using the circular view of the PATRIC Bioinformatics Resource Center (22). Functional enrichment pathways were analyzed and visualized by RAST (17), SEED (16), Database for Annotation, Visualization and Integrated Discovery, KEGG, and eggNOG (18). Functional enrichment pathways were plotted using Cytoscape (19).

## Statistical analysis

Results were presented as average ± standard error of mean in triplicate. Statistical significance was determined using Student's *t*-test for two-way comparisons. Ordinary one-way analysis of variance (ANOVA) was used in multiple comparisons. Statistical significance levels were defined as follows: $P > 0.05$ ns, $P < 0.05$*, $P < 0.01$**, and $P < 0.001$***.

## ACKNOWLEDGMENTS

This work was funded by the Swedish Research Council, Swedish Cancer Society, Radiumhemmets Research Funds, Center for Innovative Medicine, Ruth and Richard Julin Foundation, and Karolinska Institutet Foundation to M.S.C.

## AUTHOR AFFILIATIONS

[1]Division of Pathology, Department of Laboratory Medicine, Karolinska Institutet, Stockholm, Sweden

[2]Department of Dental Medicine, Karolinska Institutet, Stockholm, Sweden

[3]Department of Restorative Dental Science, College of Dentistry, King Saud University, Riyadh, Saudi Arabia

[4]Department of Diagnostics and intervention Surgery, Umeå University, Umeå, Sweden

[5]Division of Surgery, Department of Clinical Science, Intervention and Technology, Karolinska Institutet, Stockholm, Sweden

[6]Upper Abdominal Diseases, Theme Cancer, Karolinska University Hospital, Stockholm, Sweden

[7]Division of Clinical Microbiology, Department of Laboratory Medicine, Karolinska Institutet, Stockholm, Sweden

[8]Department of Clinical Microbiology, Karolinska University Hospital, Stockholm, Sweden.

[9]Department of Clinical Science, Intervention and Technology, Pancreatic Cancer Research Laboratory, Karolinska Institutet, Stockholm, Sweden

[10]Department of Cellular Therapy and Allogeneic Stem Cell Transplantation (CAST), Karolinska University Hospital Huddinge and Karolinska Comprehensive Cancer Center, Stockholm, Sweden

## AUTHOR ORCIDs

Zara Ahmad Khan http://orcid.org/0000-0002-4632-9392

Margaret Sällberg Chen http://orcid.org/0000-0002-3793-4064

## FUNDING

| Funder | Grant(s) | Author(s) |
|---|---|---|
| Swedish Cancer Foundation | 22 2406 Pj | Margaret Sällberg Chen |
| Swedish Research Council | 2020-02924 | Margaret Sällberg Chen |
| Radiumhemmets Forskningsfonder | 231312 | Margaret Sällberg Chen |

## AUTHOR CONTRIBUTIONS

Zara Ahmad Khan, Conceptualization, Investigation, Methodology, Writing – original draft, Writing – review and editing, Data curation, Formal analysis, Software, Visualization | Michał J. Sobkowiak, Investigation, Methodology, Supervision, Writing – review and editing | Mahin Ghorbani, Methodology, Resources, Writing – review and editing, Data curation, Formal analysis, Software, Validation | Tajpara Poojabahen, Investigation, Methodology, Writing – review and editing, Formal analysis | Khaled Al-Manei, Investigation, Methodology, Writing – review and editing | Asif Halimi, Conceptualization, Methodology, Writing – review and editing | Zeeshan Ateeb, Conceptualization, Methodology, Writing – review and editing | Volkan Özenci, Conceptualization, Methodology, Writing – review and editing | Rainer Heuchel, Methodology, Writing – review and editing | Matthias Löhr, Conceptualization, Resources, Writing – review and editing | Margaret Sällberg Chen, Conceptualization, Funding acquisition, Resources, Supervision, Writing – original draft, Writing – review and editing

## DATA AVAILABILITY

The Whole Genome Sequencing (WGS) data generated in this study have been deposited in the NCBI BioProject database under accession number PRJNA1171314. The transcriptomic data were published under NCBI BioProject PRJNA576932.

## ADDITIONAL FILES

The following material is available online.

### Supplemental Material

**Supplemental tables and figures (Spectrum01820-25-s0001.docx).** Tables S1 to S3, and Fig. S1 to S5.
**Graphical abstract (Spectrum01820-25-s0002.tif).** Graphical abstract.

### Open Peer Review

**PEER REVIEW HISTORY (review-history.pdf).** An accounting of the reviewer comments and feedback.

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
