## [Reviewer comments · Microbiology Spectrum]

Microbiology Spectrum

Survival mechanism of pancreatic tumor bacteria and their ability to metabolize chemotherapy drugs

Zara Khan, Michał Sobkowiak, Mahin Ghorbani, Tajpara Poojabahen, Khaled Al-Manei, Asif Halimi, Zeeshan Ateeb, Volkan Özenci, Rainer Heuchel, Matthias Löhr, and Margaret Sällberg Chen

Corresponding Author(s): Margaret Sällberg Chen, Karolinska Institutet

Review Timeline:

Submission Date:	June 12, 2025
Editorial Decision:	June 25, 2025
Revision Received:	July 12, 2025
Accepted:	July 23, 2025

Editor: Yuan Pin Hung

Reviewer(s): The reviewers have opted to remain anonymous.

Transaction Report:

DOI: <https://doi.org/10.1128/spectrum.01820-25>

Re: Spectrum01820-25 (Survival mechanism of pancreatic tumor bacteria and their ability to metabolize chemotherapy drugs)

Dear Prof. Margaret Sällberg Chen:

Thank you for the privilege of reviewing your work. Below you will find my comments, instructions from the Spectrum editorial office, and the reviewer comments.

This article is very interesting. However, the conclusions are primarily based on functional assays and whole-genome sequencing (WGS) analyses. Are there any additional in vitro or in vivo data to further support your findings?

Revision Guidelines

Sincerely,
Yuan Pin Hung
Editor
Microbiology Spectrum

Reviewer #1 (Public repository details (Required)):

The bacterial WGS data is a valuable resource for other scientists in the field.

Reviewer #1 (Comments for the Author):

Z. Ahmad Khan et al. present a comprehensive study on the bacterial diversity in IPMNs and IPMN-associated cancer, with a particular focus on bacterial interference with standard-of-care therapeutic agents such as gemcitabine and 5-FU.

Strengths of the study:

- The study convincingly demonstrates the diversity of bacterial colonization in preneoplastic pancreatic lesions.
- I appreciate the consideration of the complex pancreatic microenvironment, particularly the comparison between anaerobic and aerobic conditions (Figure 4).
- The discussion section provides candid and thoughtful insights into the limitations of the study.

Points for improvement:

- "Pancreatic cancer" is a highly heterogeneous term. Can the authors clarify whether IPMNs are precursors to a specific subtype, such as PDAC? What is the distribution? The use of PDAC cell lines and the selected chemotherapeutic agents (gemcitabine and 5-FU) suggest a PDAC context, which differs significantly from other pancreatic cancers such as insulinoma.
- Is there evidence of preferential bacterial colonization in particular pancreatic cancer subtypes? Please clarify what is meant by the "IPMN-derived invasive cancer."
- Figure 1A contains a typo in "invasive cancer."
- There is some ambiguity in Figure 2B/C regarding biological vs. technical replicates. It appears that the authors conducted two biological replicates with three technical replicates each. To strengthen the data, I would recommend including at least three independent biological replicates.
- In Figure 2D, the term "increase" relative to drug-only treatment on PDAC cells is unclear. Increase compared to what? Could this be rephrased to indicate a "rescue" or mitigation of drug effect?
- Some figures lack clarity in labeling, and in some cases, the association between labels and subfigures is ambiguous. Improved figure legends and clearer labeling would aid interpretation.

Reviewer #2 (Comments for the Author):

In this study, Khan et al. attempt to assess the relevance of bacterial metabolism on the degradation of therapeutically relevant drugs. The manuscript is well written, sufficiently easy to follow, and the underlying concept is interesting and worth exploring. The work remains mainly descriptive, and while the sequencing effort allows some speculation on the mechanisms involved in this process, there are no real mechanistic insights supported by wet lab work.

Major critique:

- 1) Because of the descriptive nature of the work, the only relevant experimental results are those described in Figure 2 and supplemented in Figure SI-2; therefore, the value of the entire manuscript relies on the data presented there. This reviewer feels that additional work and/or clarifications are required to strengthen this part of the work.
 - a. One may assume that in conditioned media, other factors besides drug degradation may give the cell lines increased survival capabilities. At a minimum, the authors need to provide experimental evidence that this is not the case. A possible experimental scenario (but not necessarily the only one) could be the following: if bacteria-conditioned media, for each relevant strain showing a statistically significant effect, was generated without drug, and added to the 5-FU or Gem controls shown on the right of each panel in 2B and 2C, would that be enough to rescue the viability? If so, this would point to factors other than drug metabolism being responsible for the observed data.

Minor critiques:

- 1) The authors show results for 4uM and 40uM. It would strengthen the experimental work to extend their analysis to a range of concentrations (such as the two fold dilutions mentioned in line 409), or at the very least a more comprehensive discussion on the therapeutic relevance of these two concentrations is needed.
- 2) The authors mention experiments being performed twice in three biological replicates. Were there technical replicates as well? Importantly, the authors should clarify what is shown in Panels 2B, 2C and the summary in Panel 2D. Is each histogram the average of the 3 biological replicates in one of the two experiments, chosen as representative?

Reviewer #3 (Public repository details (Required)):

WGS, RNA-Seq

Reviewer #3 (Comments for the Author):

Survival mechanism of pancreatic tumor bacteria and their ability to metabolize chemotherapy drugs

Overall this is a nice study focussing on an important problem, pancreatic cancer has a 9-11% chance of survival. The authors conduct a study identifying key microbes that can exist within the pancreas with the ability to prevent the mechanism of action for

two commonly used compounds used for treatment of this cancer.

Line 77, " Gut-resident microbes are increasingly known to microbial-drug interference", possible typo, cause may be added before microbial

Line 87, The significance of the following was not explained "low-or high-grade dysplasia (LGD/HGD) or associated with invasive cancer (IC)". First results section focusses on bacteria from each type of growth. For a microbiology journal it could be beneficial to explain this further in the introduction.

Line 100, Fig 1A is unnecessary, it doesn't explain anything. Instead this manuscript would benefit from a table S1 becoming main table instead of Fig 1A and 1B. The control strains are also unnecessary, comparing reference E.coli growth to different strains doesn't add anything.

Again similar to Fig 1A, Fig 1D is unnecessary as it doesn't add further information, this is due to text and Fig 1B/1C already explaining everything in Fig 1D before it is mentioned.

Line 115, Stating almost every second strain is misleading as the order they are in tables doesn't indicate this. May be better to write percentage of resistant isolates.

Line 116-122, authors state some strains are resistant or sensitive, how were these conclusions conducted. Dissimilar to antibiotics there doesn't seem to be guidelines.

Fig 1E, Table S1 and Fig S1 show nearly the same data just plotted different ways, are these all necessary.

Fig 2 and Fig S2 could be joined into a table due to the large amount of data. This data would be more beneficial to a reader as a table due to the differing concentrations and differing cell lines, a table would allow the reader to see the differences between cell lines, bacteria and compounds with greater ease.

Section "IPMN-derived Gram-negative strains are more functionally enriched compared to Gram positive 143 strains", in this analysis do tumor bacteria specifically have these changes or is this normal gram negative vs gram positive differences. Bacteria have different niches and will normally have different abundances of genes responsible for different pathways. Not sure how this adds to the study.

Line 326, "this is the first data to demonstrate that IPMN-associated microbes are influenced by their tumor environments, and they may have adapted to these environmental conditions to survive". I am unsure of the following statement. There any genome comparison of the same bacterial strains from other environments compared to the ones from this study showing differences between tumor and non-tumor origins.

Overall assessment

In this study, Khan et al. attempt to assess the relevance of bacterial metabolism on the degradation of therapeutically relevant drugs. The manuscript is well written, sufficiently easy to follow, and the underlying concept is interesting and worth exploring. The work remains mainly descriptive, and while the sequencing effort allows some speculation on the mechanisms involved in this process, there are no real mechanistic insights supported by wet lab work.

Major critiques:

- 1) Because of the descriptive nature of the work, the only relevant experimental results are those described in Figure 2 and supplemented in Figure SI-2; therefore, the value of the entire manuscript relies on the data presented there. This reviewer feels that additional work and/or clarifications are required to strengthen this part of the work.
 - a. One may assume that in conditioned media, other factors besides drug degradation may give the cell lines increased survival capabilities. At a minimum, the authors need to provide experimental evidence that this is not the case. A possible experimental scenario (but not necessarily the only one) could be the following: if bacteria-conditioned media, for each relevant strain showing a statistically significant effect, was generated without drug, and added to the 5-FU or Gem controls shown on the right of each panel in 2B and 2C, would that be enough to rescue the viability? If so, this would point to factors other than drug metabolism being responsible for the observed data.

Minor critiques:

- 1) The authors show results for 4uM and 40uM. It would strengthen the experimental work to extend their analysis to a range of concentrations (such as the two fold dilutions mentioned in line 409), or at the very least a more comprehensive discussion on the therapeutic relevance of these two concentrations is needed.
- 2) The authors mention experiments being performed twice in three biological replicates. Were there technical replicates as well? Importantly, the authors should clarify what is shown in Panels 2B, 2C and the summary in Panel 2D. Is each histogram the average of the 3 biological replicates in one of the two experiments, chosen as representative?

Survival mechanism of pancreatic tumor bacteria and their ability to metabolize chemotherapy drugs

Overall this is a nice study focussing on an important problem, pancreatic cancer has a 9-11% chance of survival. The authors conduct a study identifying key microbes that can exist within the pancreas with the ability to prevent the mechanism of action for two commonly used compounds used for treatment of this cancer.

Line 77, "Gut-resident microbes are increasingly known to microbial-drug interference", possible typo, cause may be added before microbial

Line 87, The significance of the following was not explained "low-or high-grade dysplasia (LGD/HGD) or associated with invasive cancer (IC)". First results section focusses on bacteria from each type of growth. For a microbiology journal it could be beneficial to explain this further in the introduction.

Line 100, Fig 1A is unnecessary, it doesn't explain anything. Instead this manuscript would benefit from a table S1 becoming main table instead of Fig 1A and 1B. The control strains are also unnecessary, comparing reference *E.coli* growth to different strains doesn't add anything.

Again similar to Fig 1A, Fig 1D is unnecessary as it doesn't add further information, this is due to text and Fig 1B/1C already explaining everything in Fig 1D before it is mentioned.

Line 115, Stating almost every second strain is misleading as the order they are in tables doesn't indicate this. May be better to write percentage of resistant isolates.

Line 116-122, authors state some strains are resistant or sensitive, how were these conclusions conducted. Dissimilar to antibiotics there doesn't seem to be guidelines.

Fig 1E, Table S1 and Fig S1 show nearly the same data just plotted different ways, are these all necessary.

Fig 2 and Fig S2 could be joined into a table due to the large amount of data. This data would be more beneficial to a reader as a table due to the differing concentrations and differing cell lines, a table would allow the reader to see the differences between cell lines, bacteria and compounds with greater ease.

Section "IPMN-derived Gram-negative strains are more functionally enriched compared to Gram positive 143 strains", in this analysis do tumor bacteria specifically have these changes or is this normal gram negative vs gram positive differences. Bacteria have different niches and will normally have different abundances of genes responsible for different pathways. Not sure how this adds to the study.

Line 326, "this is the first data to demonstrate that IPMN-associated microbes are influenced by their tumor environments, and they may have adapted to these environmental conditions to survive". I am unsure of the following statement. There any genome comparison of the same bacterial strains from other environments compared to the ones from this study showing differences between tumor and non-tumor origins.

Response to Reviewers

Response to editors and reviewers Re: Spectrum01820-25 (Survival mechanism of pancreatic tumor bacteria and their ability to metabolize chemotherapy drugs) by Ahmad Khan et al. submitted for publication in *Microbiology Spectrum*

Dear Yuan Pin Hung
Editor Microbiology Spectrum

Thank you for your effort and time in assessing our work. Below please find our comments to the expert reviewers.

We would like to thank the reviewers for the expert assessment. We have updated our manuscript according to the comments. Thank you once again for your expert input to further improve our manuscript. Much appreciated!

Please find our replies in **blue** and manuscript changes marked in **yellow**

Reviewer comments:

Reviewer #1 (Public repository details (Required)): The bacterial WGS data is a valuable resource for other scientists in the field.

Our reply: Thank you. Our data availability statement is now provided in Page 22.

Reviewer #1 (Comments for the Author):

Z. Ahmad Khan et al. present a comprehensive study on the bacterial diversity in IPMNs and IPMN-associated cancer, with a particular focus on bacterial interference with standard-of-care therapeutic agents such as gemcitabine and 5-FU.

Strengths of the study:

- The study convincingly demonstrates the diversity of bacterial colonization in preneoplastic pancreatic lesions.
- I appreciate the consideration of the complex pancreatic microenvironment, particularly the comparison between anaerobic and aerobic conditions (Figure 4).
- The discussion section provides candid and thoughtful insights into the limitations of the study.

Points for improvement:

- **"Pancreatic cancer" is a highly heterogeneous term. Can the authors clarify whether IPMNs are precursors to a specific subtype, such as PDAC? What is the distribution? The use of PDAC cell lines and the selected chemotherapeutic agents (gemcitabine and 5-FU) suggest a PDAC context, which differs significantly from other pancreatic cancers such as insulinoma.**

Our reply: Thank you for the advice. We have included this information in our revised manuscript in the introduction section on Page 4 and 5.

- **Is there evidence of preferential bacterial colonization in particular pancreatic cancer subtypes? Please clarify what is meant by the "IPMN-derived invasive cancer."**

Our reply: Apologies for not being clear. We have updated the requested information about the pancreatic cancer subtype and improved the wording of IPMN-derived invasive cancer. The latter is because PDAC is very common and is an invasive cancer, the terms "invasive cancer in the pancreas" and "PDAC" are often used interchangeably in clinical practice.

This updated information is now provided in Page 4 and 6.

- Figure 1A contains a typo in "invasive cancer."

Our reply: thank you, the typo has been fixed.

- There is some ambiguity in Figure 2B/C regarding biological vs. technical replicates. It appears that the authors conducted two biological replicates with three technical replicates each. To strengthen the data, I would recommend including at least three independent biological replicates.

Our reply: We sincerely thank the reviewer for the observation and the opportunity to clarify the experimental design and reproducibility. Initially, we conducted multiple pilot studies to ensuring the consistency and optimized our experimental protocol. Based on the optimal protocol, we then performed two independent experiments, each consisting of three biological replicates for the given condition. These were then combined to form a single dataset with a total of six biological replicates. We would like to emphasize that no technical replicates were included in this study; all six replicates represent independent biological samples. By optimizing our protocol and normalizing our data, we could control the variability and ensure the robustness of our result. We apologize for not making this point sufficiently clear in the original manuscript and have now clarified this in the revised version.

We have improved the clarity in our updated figure legend for Figure 2B/C.

- In Figure 2D, the term "increase" relative to drug-only treatment on PDAC cells is unclear. Increase compared to what? Could this be rephrased to indicate a "rescue" or mitigation of drug effect?

Our reply: We sincerely thank the reviewer for their valuable comment and apologize for the lack of clarity in this figure. The term "increase," was intended to convey the change of cell viability in relation to drug alone i.e. controls (without exposure to bacterial strains). We fully agree with the reviewer's suggestion and have updated the label to "% of viability increase (relative to drug control)". This change has been incorporated into the revised manuscript. We are grateful to the reviewer for this insightful observation, which has helped to improve both the clarity and overall quality of the manuscript.

We have provided corrected Figure 2D.

- Some figures lack clarity in labeling, and in some cases, the association between labels and subfigures is ambiguous. Improved figure legends and clearer labeling would aid interpretation.

Our reply: Apologies for not being clear. We have now double-checked all figure labels and made legends consistent and aligned to details in our revised manuscript.

Reviewer #2 (Comments for the Author):

In this study, Khan et al. attempt to assess the relevance of bacterial metabolism on the degradation of therapeutically relevant drugs. The manuscript is well written, sufficiently easy to follow, and the underlying concept is interesting and worth exploring. The work remains mainly descriptive, and while the sequencing effort allows some speculation on the mechanisms involved in this process, there are no real mechanistic insights supported by wet lab work.

Major critique:

1) Because of the descriptive nature of the work, the only relevant experimental results are those described in Figure 2 and supplemented in Figure SI-2; therefore, the value of the entire manuscript relies on the data presented there. This reviewer feels that additional work and/or clarifications are required to strengthen this part of the work.

a. One may assume that in conditioned media, other factors besides drug degradation may give the cell lines increased survival capabilities. At a minimum, the authors need to provide experimental evidence that this is not the case. A possible experimental scenario (but not necessarily the only one) could be the following: if bacteria-conditioned media, for each relevant strain showing a statistically significant effect, was generated without drug, and added to the 5-FU or Gem controls shown on the right of each panel in 2B and 2C, would that be enough to rescue the viability? If so, this would point to factors other than drug metabolism being responsible for the observed data.

Our reply: We sincerely thank the reviewer for their thoughtful and detailed observation. Indeed, we also have considered this in our initial studies and checked if *bacteria-conditioned media, for each relevant strain, generated without drug were enough to rescue the viability.*”. In those pre-tests, we did not see significant rescue of the cell viability following incubation with any of the bacteria-condition media prepared without the drug (Figure below). Our data therefore helped to exclude factors other than drug metabolism could be responsible for the observed data. We appreciate the reviewer’s suggestion, and we hope this clarifies our rationale for its exclusion from the main manuscript. A note has been included in our revised manuscript (Page 7).

Figure note: PANC-1 cell viability following 48 hours exposure to the indicated bacterial-conditioned medium, expressed in relation to the viability in PANC-1 culture medium.

Minor critiques:

1) The authors show results for 4uM and 40uM. It would strengthen the experimental

work to extend their analysis to a range of concentrations (such as the two fold dilutions mentioned in line 409), or at the very least a more comprehensive discussion on the therapeutic relevance of these two concentrations is needed.

Our reply: We sincerely thank the reviewer for their thoughtful suggestion and the opportunity to clarify our rationale for selecting the drug concentrations used in this study. At the initial phase of the project, we used two-fold serial dilutions to explore a wide range of concentrations up to 1000 μM . At higher concentrations, the cell toxicity was more drastic. Based our literature review, we ultimately chose 4 μM and 40 μM for our main experiments to remain within a physiologically relevant range, as published in literature below:

For example, Spanogiannopoulos et al. (2022, *Nature Microbiology*, PMID: PMC9530025, DOI: 10.1038/s41564-022-01226-5) used 10 μM 5-FU in vitro when investigating microbiota-driven modulation of drug activity. Similarly, LaCourse et al. (2022, *Cell Reports*, DOI: 10.1016/j.celrep.2022.111625) used 10 $\mu\text{g/mL}$ of 5-FU, equivalent to $\sim 77 \mu\text{M}$, to explore microbial metabolism of 5-FU. In the case of gemcitabine, Geller et al. (2017, *Science*, DOI: 10.1126/science.aah5043) used 50 μM in their PANC-1/microbial-drug interaction experiments.

We appreciate the reviewer's comment, which has helped us clarify this point. We have included this clarification and the supporting citations in our revision accordingly (Page 7).

2) The authors mention experiments being performed twice in three biological replicates. Were there technical replicates as well? Importantly, the authors should clarify what is shown in Panels 2B, 2C and the summary in Panel 2D. Is each histogram the average of the 3 biological replicates in one of the two experiments, chosen as representative?

Our reply: We thank the reviewer for their valuable observation and the opportunity to clarify the reproducibility and structure of the experiment. To establish a robust and reproducible experimental protocol, we initially conducted several pilot experiments. These preliminary studies were essential for optimizing conditions and ensuring consistency across replicates. Following protocol optimization, we performed two independent experiments, each consisting of three biological replicates. The data from these experiments were then combined to form a single experimental panel comprising a total of six biological replicates. We would like to emphasize that no technical replicates were included in this study; all six replicates represent independent biological samples. We apologize for not making this point sufficiently clear in the original manuscript. In response to the reviewer's suggestion, we have updated the legends of Figure 2B and 2C to better reflect the experimental design and clarify the number of biological replicates used. These revisions have been incorporated into the revised manuscript.

Reviewer #3 (Public repository details (Required):
WGS, RNA-Seq (update)

Our reply: Our data availability statement is now provided in Page 22

Reviewer #3 (Comments for the Author):

Survival mechanism of pancreatic tumor bacteria and their ability to metabolize chemotherapy drugs

Overall this is a nice study focussing on an important problem, pancreatic cancer has a 9-11% chance of survival. The authors conduct a study identifying key microbes that can exist within the pancreas with the ability to prevent the mechanism of action for two commonly used compounds used for treatment of this cancer.

Line 77, " Gut-resident microbes are increasingly known to microbial-drug interference", possible typo, cause may be added before microbial

Our reply: Apologies, the typo has been corrected.

Line 87, The significance of the following was not explained "low-or high-grade dysplasia (LGD/HGD) or associated with invasive cancer (IC)". First results section focusses on bacteria from each type of growth. For a microbiology journal it could be beneficial to explain this further in the introduction.

Our reply: Thank you for raising this point. We have improved the description of low-or high-grade dysplasia (LGD/HGD) or associated with invasive cancer (IC).

This is now provided in Page 5.

Line 100, Fig 1A is unnecessary, it doesn't explain anything. Instead this manuscript would benefit from a table S1 becoming main table instead of Fig 1A and 1B. The control strains are also unnecessary, comparing reference E.coli growth to different strains doesn't add anything. Again similar to Fig 1A, Fig 1D is unnecessary as it doesn't add further information, this is due to text and Fig 1B/1C already explaining everything in Fig 1D before it is mentioned.

Our reply: We appreciate reviewer's comment and constructive advice. We have incorporated Table S1 key content to replace previous Figure 1A, to improve the presentation of our data. We agree the updated Figure 1 is better and allows readers to view the relevant strain information directly within the main figure, without consulting supplementary materials. Regarding the other sub-figures, we respectfully believe that these panels provide important information. Specifically, they provide key phenotypic responses and schematic presentation of the concept. We have revised and toned down these items by re-arrangement of their order in the panel. Regarding the reference *E. coli* strain, we believe it offers values by serving as a technical reference to allow our interpretation of differential responses across the strains. Therefore, we opt to keep it as an effort to reach completeness.

Line 115, Stating almost every second strain is misleading as the order they are in tables doesn't indicate this. May be better to write percentage of resistant isolates.

Our reply: Thank you for the advice. This has been changed to percentage. (Page 6)

Line 116-122, authors state some strains are resistant or sensitive, how were these conclusions conducted. Dissimilar to antibiotics there doesn't seem to be guidelines.

Our reply: We thank the reviewer for their insightful comment. It is true that guideline is yet not available. Meanwhile, our classification of bacterial strains as “sensitive” or “resistant” was based on results obtained from a dose–response analysis using two-fold serial dilutions of the chemotherapeutic drugs 5-fluorouracil (5-FU) and gemcitabine. We began with a starting concentration of 1000 μ M and performed nine two-fold serial dilutions. Based on the resulting dose–response curves, we calculated the IC₅₀ values for each strain. Strains with an IC₅₀ that fell within the tested concentration range were categorized as sensitive, while those with an IC₅₀ exceeding the highest concentration tested were classified as resistant.

This information has been added to improve our method description (provided in Page 19)

Fig 1E, Table S1 and Fig S1 show nearly the same data just plotted different ways, are these all necessary.

Our reply: Thank you for the comment. While the processed data is given in a heat map in Figure 1E, to allow readers overall assessment on the effects of 5-FU and gemcitabine across all strains. The dataset on the strain-specific dose–response curves are provided in Figure S1 and Table S2 as supportive material for readers to ensure transparency and data quality assessment. This has been requested previously in our publications therefore we made this effort to include these items, which generally allow better interpretation of our work. We hope for the reviewer’s understanding.

Fig 2 and Fig S2 could be joined into a table due to the large amount of data. This data would be more beneficial to a reader as a table due to the differing concentrations and differing cell lines, a table would allow the reader to see the differences between cell lines, bacteria and compounds with greater ease.

Our reply: We thank the reviewer for this suggestion. Indeed, we tried to combine these two items. However, due to the already extensive number of figure panels and the use of various assays, the combined figure became rather difficult to read and resulted in a loss of clarity. Therefore, we chose to present Figures 2 and S2 separately to avoid confusion.

Section "IPMN-derived Gram-negative strains are more functionally enriched compared to Gram positive 143 strains", in this analysis do tumor bacteria specifically have these changes or is this normal gram negative vs gram positive differences. Bacteria have different niches and will normally have different abundances of genes responsible for different pathways. Not sure how this adds to the study.

Our reply: We appreciate the reviewer’s insightful comment. Indeed, our Gram-negative strains also belong to the *Gammaproteobacteria* phylum, known to include clinically significant pathogens. These genomes have been more extensively studied than those of Gram-positive strains, contributing to the greater functional annotation and pathway-level insights currently available for Gram-negative bacteria. However, we think it is important to demonstrate that our tumor-associated bacterial strains are not an exception. Given the typical features of Gram-negative bacteria - greater phenotypic and genotypic diversity and more complex cell wall structure, these strains may possess broader functional capabilities.

Another point of interest in this section, shown in Figure 3, highlights an example involving *E. cloacae* strains isolated from tumors of different malignancy grades (low-grade dysplasia,

high-grade dysplasia, and invasive cancer). Despite being the same species, these strains exhibited distinct gene expression profiles across different pathways, suggesting genomic and phenotypic differences not previously reported.

In line with the reviewer's comment, we are pursuing a related project that focuses specifically on *E. cloacae* bacteria strains isolated from different anatomical regions. We appreciate the reviewer drawing attention to this currently unresolved direction. A more in-depth evaluation of this topic should provide valuable insights and potential clinical implications in the near future.

We have improved our Discussion with a summary of this feedback in Page 15.

Line 326, "this is the first data to demonstrate that IPMN-associated microbes are influenced by their tumor environments, and they may have adapted to these environmental conditions to survive". I am unsure of the following statement. There any genome comparison of the same bacterial strains from other environments compared to the ones from this study showing differences between tumor and non-tumor origins.

Our reply: We are very grateful to the reviewer for this thoughtful comment which we completely agree. We are now conducting a follow-up study that specifically focuses on comparing *E. cloacae* strains isolated from tumor versus non-tumor regions. Since the current manuscript focuses on broader microbial-drug interactions, our new study aims separately to address the challenge highlighted in the reviewer's comment. In line with this comment, we have now toned down the statement in line 326 and elaborated on the need to explore this direction in future work accordingly (Revised in Page 15)

Re: Spectrum01820-25R1 (Survival mechanism of pancreatic tumor bacteria and their ability to metabolize chemotherapy drugs)

Dear Prof. Margaret Sällberg Chen:

Your manuscript has been accepted, and I am forwarding it to the ASM production staff for publication. Your paper will first be checked to make sure all elements meet the technical requirements. ASM staff will contact you if anything needs to be revised before copyediting and production can begin. Otherwise, you will be notified when your proofs are ready to be viewed.

Sincerely,
Yuan Pin Hung
Editor
Microbiology Spectrum

Reviewer #2 (Public repository details (Required)):

WGS datasets

Reviewer #2 (Comments for the Author):

1) Would recommend adding the conditioned media experiment, shown in the response to the reviewer, to the supplemental data; supplemental data are precisely meant to include that kind of data and controls. 2) Would recommend specifying not only that "Experiments were repeated twice with three independent biological replicates each" but also that each histogram in the figure shown represent the average of the 6 replicates.

Reviewer #3 (Public repository details (Required)):

RNA-Seq & WGS

Reviewer #3 (Comments for the Author):

The authors have supplied an updated manuscript with clarifications regarding reviewers comments